# Renal Disturbances during and after Radioligand Therapy of Neuroendocrine Tumors—Extended Analysis of Potential Acute and Chronic Complications

**DOI:** 10.3390/ijms24087508

**Published:** 2023-04-19

**Authors:** Marek Saracyn, Adam Daniel Durma, Barbara Bober, Arkadiusz Lubas, Maciej Kołodziej, Waldemar Kapusta, Beata Dmochowska, Grzegorz Kamiński

**Affiliations:** 1Department of Endocrinology and Radioisotope Therapy, Military Institute of Medicine—National Research Institute, 04-141 Warsaw, Poland; adurma@wim.mil.pl (A.D.D.);; 2Department of Internal Diseases, Nephrology and Dialysis, Military Institute of Medicine—National Research Institute, 04-141 Warsaw, Poland

**Keywords:** neuroendocrine neoplasm (NEN), radioligand therapy (RLT), renal, acute, chronic, complications, DTPM, IL-18, KIM-1, ^177^Lu, ^90^Y, fractional calcium excretion

## Abstract

Neuroendocrine tumors (NEN) are a group of neoplasms that arise from hormonal and neural cells. Despite a common origin, their clinical symptoms and outcomes are varied. They are most commonly localized in the gastrointestinal tract. Targeted radioligand therapy (RLT) is a treatment option which has proven to be successful in recent studies. However, the possible outcomes and true safety profile of the treatment need to be fully determined, especially by new, more sensitive methods. Our study aimed to present an extended analysis of acute and chronic renal complications during and after radioligand therapy using, for the first time in the literature, innovative and complex renal parameters. Forty patients with neuroendocrine tumors underwent four courses of radioligand therapy with [^177^Lu]Lu-DOTATATE or [^177^Lu]Lu/[^90^Y]Y-DOTATATE. Radioisotopes were administrated in intervals of 8–12 weeks, with concurrent intravenous nephroprotection. New detailed and sensitive renal parameters were used to determine the renal safety profile during and after radioisotope therapy for standard treatment of NEN. During the first and fourth courses of RLT, no change in the glomerular filtration rate (GFR) was observed. However, long-term observations one year after the treatment showed a 10% reduction in the GFR. During the first course of treatment, the fractional urea and calcium excretions increased, while the fractional potassium concentration decreased. The fractional calcium excretion remained highly increased in long-term observations. Decreases in urine IL-18, KIM-1 and albumin concentrations were observed during RLT. The concentrations of IL-18 and KIM-1 remained low even a year after therapy. The ultrasound parameters of renal perfusion changed during treatment, before partially returning to the baseline one year after therapy, and were correlated with the biochemical parameters of renal function. A permanent increase in diastolic blood pressure was correlated with the decrease in the GFR observed during the study. In this innovative and complex renal assessment during and after RLT, we found a permanent 10% per year decrease in the GFR and noticeable disturbances in renal tubule function. The diastolic blood pressure also increased.

## 1. Introduction

Neuroendocrine neoplasms (NENs) are a group of tumors arising from hormonal and neural cells. Despite a common origin, their symptomatology and potential complications are different. Their most common location is the gastrointestinal tract [1,2]. They are usually “non-functioning” tumors and present no symptoms until mass effect or liver metastases are detected. Some tumors, described as “functioning” (such as gastrinoma or insulinoma), have hormonal activity, and due to specific symptomatology can be diagnosed earlier [3,4].

Peptide receptor radionuclide therapy (PRRT), lastly defined as targeted radioligand therapy (RLT), has recently been gaining increasing acceptance and appears to be an efficient and tolerable option for NEN treatment. This therapy is recommended for grade G1 or G2 NEN, and in some cases of G3 with confirmed somatostatin receptor expression in ^68^Ga-PET or 99mTc scintigraphy [4]. Nowadays, the radioisotopes used for treatment are ^177^lutetium alone or ^177^lutetium combined with ^90^yttrium as a tandem therapy [5]. ^177^Lutetium is characterized by an over four times lower maximum energy of beta emission and a five times shorter range of activity than ^90^yttrium. Thanks to these features, ^177^lutetium is theoretically less toxic and should result in fewer complications. The most commonly observed adverse events (AEs) are renal, liver and bone marrow dysfunction, which could lead to disqualification from or discontinuation of therapy [6].

Due to the renal excretion of radioisotopes used in RLT, the kidneys are the most vulnerable to radiation; thus, renal adverse events are often described in the literature. The number of nephrons in human kidneys is determined from birth, so their loss due to any injury leads to an irreparable decline in renal function [7]. Moreover, [^177^Lu]Lu-DOTA-TATE or [^90^Y]Y-DOTA-TATE are reabsorbed in proximal tubules by the endocytic megalin/cubilin receptor complex, resulting in the retention of radionuclides in the renal interstitium. There is also a small, yet noticeable, expression of the somatostatin receptor in the kidneys, which can increase the unfavorable effects of RLT on local tissues [8]. Thus, the radioisotope dose is limited mainly by possible renal complications. Previous data have shown an average annual decrease in the glomerular filtration rate after RLT with [^177^Lu]Lu-Octreotate of about 3% (with no patient showing a decrease of more than 20%). Nevertheless, the study was conducted on a group with initially normal kidney functions (GFR 108 ± 5 mL/min) [9]. There are several ways to overcome the nephrotoxicity of RLT, of which the most important is nephroprotection with amino acid infusion and good hydration of the patient [10]. The possible use of renin-angiotensin-aldosterone system blockers (angiotensin-converting enzyme inhibitors, angiotensin II receptor blockers or aldosterone receptor blockers), α1-microglobulin, amifostine or even metformin requires further investigation [8].

Year-on-year, the development of diagnostic methods leads to the discovery of new molecules that could be helpful in assessing kidney injuries. The most recently discovered biomarkers describing glomerular injury are nephrin, podocalyxin, podocin, netrin-1 or pyruvate kinase M2. Moreover, kidney injury molecule 1 (KIM-1), interleukin 18 (IL-18), neutrophil gelatinase-associated lipoocalcin (NGAL) or insulin-like growth factor binding protein 7 (IGFBP7) and urinary tissue inhibitor of metalloproteinases-2 (TIMP-2) are parameters of tubular and interstitial disorders [11].

There are also new biomarkers used for assessing chronic kidney disease and predicting its outcomes, such as asymmetric dimethylarginine (ADMA), symmetric dimethylarginine (SDMA), uromodulin and RNA fragments (miRNA, ncRNA and lincRNAs), as well as acute injury KIM-1 and NGAL [12].

Thus, the aim of our study was to perform an extended analysis of acute and chronic renal complications during and after radioligand therapy. For the first time in the literature, we used innovative and complex renal function parameters to evaluate renal filtration, tubules and the renal endothelium function.

## 2. Results

The clinical and epidemiological data of the study group are presented in Table 1.

### 2.1. Acute Complications

During course I, there were no differences in the glomerular filtration rate observed in the study group. None of the analyzed factors such as age, gender, body mass index (BMI), other chronic diseases, previous administered chemotherapy or primary tumor location influenced these results. Among patients receiving tandem therapy compared to ^177^lutetium alone, a small but significant increase in GFR was observed (^177^Lu/^90^Y: Δ = 6.64; SD = 10.38; ^177^Lu: Δ = –2.62; SD = 0.64; *p* = 0.018). Moreover, the results showed an increase in fractional urea excretion (*p* = 0.001*)* and urine pH (*p* = 0.047) with a decrease in fractional potassium excretion (*p* = 0.044*),* urine IL-18 concentration (*p* < 0.001) and urine albumins (*p* = 0.004). Again, no demographic and clinical factors influenced the results.

During course IV, there were also no differences in GFR; however, in a subgroup of the patients without diabetes mellitus (diagnosed before treatment), a significant decrease in GFR was observed (*p* = 0.045). In addition, increases in fractional calcium and urea excretion were noted (*p* = 0.046 and *p* < 0.001, respectively), while decrease in albuminuria (*p* = 0.035) and the albumin/creatinine ratio (ACR) (*p* = 0.003) were also found. Again, the results did not correlate with gender, age, BMI, comorbidities, primary tumor location or the type of radioligand used. Detailed data are presented in Table 2.

Comparing the results obtained during course I and IV, there was a more intense decrease in GFR (*p* = 0.036) and an increase in fractional potassium excretion (*p* = 0.012). Furthermore, smaller decreases in albuminuria (*p* = 0.02) and urine IL-18 (*p* < 0.001) after course IV were also found (Table 3 and Table 4).

### 2.2. Chronic Complications

Chronic renal complication was assessed thrice by comparing results obtained before course I and course IV (Analysis One), before course IV and a year after course IV (Analysis Two) and before course I and a year after course IV (Analysis Three). The summarized results are shown placed in Table 5.

In Analysis One (course I vs. IV), we observed decreases in KIM-1 (*p* = 0.003), IL-18 (*p* < 0.001) and albumin (*p* = 0.011) concentrations. A larger decrease in KIM-1 was observed in patients receiving tandem therapy (*p* = 0.012). A more significant IL-18 decrease was noted among patients with diabetes (n = 12, *p* = 0.003*)* receiving tandem therapy (*p* = 0.013). Comorbidities and previous chemotherapy did not influence these results.

Analysis One also included an assessment of ultrasound dynamic tissue perfusion measurements (DTPM) in the renal cortex. We observed that the vascular area of renal blood vessels was significantly larger before course IV compared to course I (*p* = 0.031). In addition, the renal cortex perfusion (RCP) also statistically increased (*p* = 0.038) (Table 6).

In Analysis Two (course IV vs. follow-up 1 year after the RLT), we observed a decrease in GFR at the statistical trend level (*p* = 0.06). However, after changing the GFR calculation formula (MDRD instead of CKD-EPI), statistical significance was reached (*p* = 0.024) (Table 7). Analysis of chronic complications depending on radioisotope used was presented in Table 8.

A larger decrease in GFR was observed in overweight patients (BMI ≥ 25 kg/m^2^) (*p* = 0.027) and those with hypercholesterolemia (*p* = 0.002). No other demographic or medical factors affected the results. Moreover, a year after the treatment, we observed an increase in fractional calcium urine excretion (*p* = 0.003), which was more significant in patients receiving tandem therapy (*p* < 0.029) and those with diabetes (*p* = 0.003).

Analysis Two also covered the DTPM ultrasound examination (before course IV vs. a year after therapy). Unlike Analysis One (course I vs. IV), decreases in the vascular area (*p* = 0.025) and renal cortex perfusion (*p* = 0.013) were observed. No other measured parameters reached statistical significance (Table 6).

In Analysis Three (course I vs. follow-up one year after RLT), a decrease in GFR with a simultaneous increase in creatinine was observed (*p* = 0.009 and *p* = 0.036, respectively). Other demographic or medical factors, except the primary tumor location, did not affect the results significantly. Nevertheless, the extrapancreatic tumor location was associated with a more notable decrease in GFR (*p* = 0.031). Chemotherapy previous to RLT did not significantly affect the results; however, among these patients, a larger decrease in GFR was noticed (Δ = −13.24 mL/min/1.73 m^2^ vs. Δ = −5.53 mL/min/1.73 m^2^; *p* = 0.247). Moreover, decreases in urine IL-18 and KIM-1 concentrations were observed a year after the treatment, but the results were only at the level of the statistical trend (*p* = 0.084 and *p* = 0.155, respectively). Furthermore, an almost six-fold increase in fractional calcium excretion remained persistent in this phase of the study (*p* = 0.009). This phenomenon was not observed for any other analyzed ions or particles (Table 5).

Analysis Three showed that the DTPM parameters measured a year after therapy partially returned to levels before treatment. However, comparing measurements from course I with a year after therapy, the mean diastolic blood pressure increased significantly from 83.17 ± 13.52 to 90.83 ± 14.50 mmHg (*p* = 0.048). This significant change in diastolic blood pressure correlated with a 10% decrease in GFR in this period (Table 5). VA and RCP levels in long-term observations also decreased, but not significantly (*p* = 0.397 and *p* = 0.213, respectively) (Table 6).

Additionally, data of RCP and DBP were summarized as a box plot during the whole time of observation and are presented in Figure 1 and Figure 2.

## 3. Discussion

In our study, for the first time in the literature, we used innovative and complex parameters to assess the renal safety profile after radioligand therapy of NEN. Some of these parameters had never been used before for such an evaluation; thus, an accurate data comparison and discussion is limited.

In the study, the glomerular filtration rate was assessed by the most updated CKD-EPI formula, as it better describes renal filtration, especially in lower stages (G1 and G2) of chronic kidney disease [13,14,15]. We have found that the first administration of radioisotope therapy resulted in no change in glomerular filtration in the whole study group. However, in the subgroup that received tandem therapy, a surprising increase in GFR was observed. These results may arise from differences in baseline GFR and creatinine concentrations in both subgroups (GFR_177-Lu_ = 82.06 mL/min/1.73 m^2^ vs. GFR_177-Lu/90-Y_ 94.5 mL/min/1.73 m^2^), as well as from a hemodynamic autoregulatory upregulation effect in renal glomeruli. The second option is more probable, as the results correlated with a higher energy of the radionuclide used. Comparing the renal parameters before the fourth course and a year after therapy, we noticed a GFR decrease at a statistical trend level. Significance was reached by changing the GFR calculation formula (MDRD instead of CKD-EPI), although the most important finding was that continuous GFR loss was seen in almost all patients treated with RLT. Long-term observations (a year after the last course versus the first course) showed a persistent decrease in the glomerular filtration rate—almost 10% compared to baseline values. This value of GFR reduction is relatively high considering a normal population. In a healthy population over 40 years old, the average annual decrease in GFR is estimated to be about 1% [16,17]. German data advocate for an approximate loss of 1 mL/min/m^2^, even beyond the age of thirty [18]. Demographical or medical factors that affected GFR during observation were the tumor location (other than the pancreas), obesity/overweight and hypercholesterolemia. It is well known that obesity correlates with a higher prevalence of dyslipidemia, so the obesity itself could be the real factor correlating the results [19,20]. A possible explanation for tumor location influence could be the shorter distance from the tumor (or its metastasis) to the kidneys, which is connected with higher doses of radiation. In our study, chemotherapy previous to RLT did not statistically influence the GFR decrease. On the other hand, other authors have observed that previous chemotherapy could be related to a higher rate of renal AEs, due to the potential nephrotoxic effect of cytostatics. The “cumulate damage” caused by both lines of treatment can accelerate the GFR loss and the serum creatinine increase [21,22,23,24].

Previous studies have confirmed that a decrease in GFR in patients treated with RLT is common, but grading of adverse events remains reasonable and acceptable [25,26]. Serious renal AEs due to RLT are still considered rare [27], and in long-term observations, the benefits from the treatment exceed the potential risks [28,29]. Although, as we mentioned above, a 10% reduction in the annual filtration rate is a rather high value compared to the healthy population. Though the tandem therapy appeared to be relatively safe, and data from the literature have confirmed low numbers of adverse events and its acceptable grading, RLT with the mixture of ^177^Lu/^90^Y can theoretically cause a larger GFR decrease than ^177^Lu alone [30,31]. The higher range and maximum energy of ^90^Y, despite its higher potential to cause AEs, can simultaneously bring a higher level of overall survival [6,32]. Interestingly, an additional analysis of subgroups of patients with different baseline GFR levels in our study showed an almost four-times higher decrease in renal filtration in long-term observations among patients with a GFR below 60 mL/min/1.73 m^2^ (${∆}_{GFR>60\;mL/min/1.73\;m^{2}}$ = −4.76 vs. ${∆}_{GFR<60\;mL/min/1.73\;m^{2}}$ = −16.50). However, the results were not statistically significant. In conclusion, the potential influence on the kidneys needs to be taken into consideration before qualification for therapy, especially in patients with a low baseline GFR.

During the first course of RLT, we also observed significant decreases in fractional potassium and urea excretion. An increase in urine pH was also noticed. During course IV, we found significant increases in fractional calcium and urea excretion. In the control a year after the treatment, we observed an even higher increase in calcium excretion. The value of this parameter was almost six times higher and was more significant in patients with diabetes and those who received tandem therapy. This phenomenon was observed for the first time in the literature during and after the RLT, and it was not significant for any other fractionally excreted ions or particles. Changes in urine pH, potassium and urea excretion were not found during course IV, so the potential influence of amino acid nephroprotection can be eliminated. This is probably due to a decreased bicarbonate (HCO3^−^) absorption in the proximal tubule, leading to a consequent increase in urinary bicarbonate excretion and elevating the urine pH [33].

Kanbara et al. tried to estimate the influence of alkaline and acidic diets on urine pH, and noted that the dysfunction of renal tubules was correlated with uric acid excretion and consequently urine pH [34]. Naqvi et al. evaluated the fractional excretion of sodium, potassium and magnesium to determine its value in cyclosporin (CsA)-induced nephrotoxicity in renal allograft transplant recipients. They proved that, among those parameters, only magnesium excretion correlated with CsA-induced nephrotoxicity [35]. The fractional excretion of electrolytes was also assessed in studies concerning chemotherapy-induced nephrotoxicity [36,37]. Ho et al. showed significant increases in the renal fractional excretions of calcium and glucose. The calcium parameters returned to baseline before the next ifosfamide treatment cycle, while glucose excretion remained high [36]. Fractional element excretion and urine pH parameters in our study were assessed to determine the potential influence of RLT on renal tubule disfunction. The observed correlations, although still not fully explained and requiring further investigations, indicated the potential treatment effects. It is worth noting that almost all the results did not correlate with any demographic or medical factors such as gender, age, BMI or primary tumor location, so this fact can be useful in extending group of patients that can benefit from RLT [38]. The most important conclusion is that fractional calcium excretion can become a predictor of potential nephrotoxicity, especially in patients with urolithiasis or even only with its risk factors. No previous studies have focused on an evaluation of the described parameters; thus, further studies on larger groups of patients are necessary.

Our results also showed that the urine IL-18 concentration decreased significantly during the first course of treatment and remained low in the long-term observations. Interleukin-18 is a modern marker of inflammatory processes, mainly in the renal interstitium. It describes processes ongoing in deep layers of renal tissues [39,40]. The results showed that processes which started during the first course of RLT were persistent a year after treatment, but the results were not expected. We also observed that the urine KIM-1 concentration decreased during the first course of treatment (not significantly), and the concentration remained low in the long-time observations. Kidney Injury Molecule-1 is a protein antigen located in renal tubule cells. Its increased value indicates damaging processes in renal tubules and can help to determine the kidney injury origin. The parameter was chosen as previously described as a sensitive marker of acute tubular and kidney injuries [41,42]. However, the KIM-1 decrease correlated with a GFR decrease as well as an IL-18 decrease; thus, the results suggest that these parameters could not be used in assessing kidney injuries induced by RLT.

When designing our study, we assumed that IL-18 or KIM-1 concentrations could be good markers of nephrotoxicity during and after RLT, just as they are for toxic or ischemic kidney injuries [41,42,43,44]. However, at all time points of the study, the concentrations of both molecules were decreased. We assume that the observed changes may have resulted from an immunosuppressive effect of the radioisotope in the kidneys. Both ^177^lutetium and ^90^yttrium, despite being therapeutically used for their beta-minus emissions, also emit gamma radiation. These emissions can affect local tissue, especially of the kidneys and their filtration and excretion functions [45,46]. The immunomodulatory effect of gamma radiation on human tissues is known, so the presented hypothesis of RLTs effect on KIM-1 and IL-18 concentrations arises from the features of these radioisotopes [47,48,49,50]. Another explanation of RLTs effect on KIM-1 and IL-18 originates from the potential inhibition of pro-inflammatory protein synthesis. In work by Bogdándi, mice irradiated with an external source of gamma radiation presented decreases in the expressions of T helper 1 and 2 (Th1 and Th2) lymphocytes and their cytokines, which were dependent on radiation dose (cytokines decreased after low doses and increased after high doses). All effects of intravenously administrated sources of radiation (beta-minus + gamma emission for ^177^Lu or ^177^Lu/^90^Y) and their possible local effects are still ambiguous [51]. Thus, the role of KIM-1 and IL-18 concentrations regarding RLT nephrotoxicity need to be determined in future studies.

In our study, the concentration of urine albumin also unexpectedly decreased during the first course of treatment for the whole duration of RLT (the first vs. the fourth course), and returned to the baseline levels a year after treatment. Albuminuria is a well-known, sensitive marker of glomerular dysfunction, so the effect of RLT, which could potentially damage the renal glomeruli, was surprising and completely opposite to expected [52]. Reduced albuminuria during treatment may result from the effect of radioisotopes on the glomerular filtration membrane—especially the endothelial cells and podocytes. It is possible that the radioisotope affects the charge of the filtration membrane (increasing its negative charge), thus “sealing” it. The other possibility is that a decreased albumin filtration results from a higher renal blood flow, similar to GFR mentioned above.

Renal perfusion measured with ultrasonography showed increases in the vascular area of renal blood vessels and renal cortex perfusion during therapy, but the parameters did normalize partially a year after therapy. The perfusion increase in the renal cortex during treatment was not caused by systemic hemodynamic changes and did not correlate with any observed results, so vasodilation of the renal cortex was probably caused by radioisotope flow and its activity. The described phenomenon could correspond to both the use of renal reserves for elimination of the radioisotope or temporary dysfunction of the hemodynamic autoregulation of renal cortex glomerular vessels. Furthermore, we also found increased diastolic blood pressure a year after therapy. Garofalo et al. in their metanalysis stated that hypertension and blood pressure can be independent predictors of decreased GFR levels in the general population [53]. Additionally, vice versa, a progressive decrease in GFR during the course of different kidney diseases inevitably leads to an increase in systemic blood pressure. It is well known that glomerulonephritis, or diabetic kidney disease, can lead to an increase in systemic blood pressure [54,55,56,57,58]. Moreover, the reduced area of the renal cortex arteries estimated in DTPM can help to differentiate between hypertensive nephropathy and glomerulonephritis [59]. In our study, long-term observations proved that the increase in DBP was correlated with the decrease in GFR, and a possible mechanism of this is the loss of active nephrons and a proper renal filtrate function [60,61]. GFR was also correlated with the area of renal vessels and renal cortex perfusion in DTPM; thus, DTPM could be used in future assessments of kidney function during and after RLT. Further studies are necessary to establish its true value.

### 3.1. Study Limitations

The low number of patients who took part in the follow-up one year after therapy, mainly due to the ongoing COVID-19 pandemic, was the biggest limitation. Moreover, a lack of consent from all patients treated with RLT in our department limited the number of patients in the study group.

### 3.2. Study Strengths

The main advantages of this study were the prospective character of the study, the protocol constructed to analyze renal complications and the attempt to establish new laboratory and ultrasonography parameters that could be useful in assessing kidney injury and dysfunction. Such large numbers of innovative complex renal parameters assessing both renal filtration and tubular and endothelial functions have not been analyzed in previous studies.

## 4. Materials and Methods

### 4.1. Patients and Protocol

The study was performed from 2017 to 2020 in a group of 40 patients with neuroendocrine tumors, who were hospitalized in the Endocrinology and Radioisotope Therapy Department of Military Institute of Medicine, National Research Institute, Warsaw, Poland. The study was conducted within the guidelines of the Helsinki Declaration and approved by the local Bioethical Committee (52/WIM/2017).

After being qualified for RLT, all patients consented to take part in the study. Intravenous administration of 7.4 GBq (200 mCi) [^177^Lu]Lu-DOTATATE was performed in 30 patients. Tandem therapy of 50 mCi (1.85 GBq) [^177^Lu]Lu-DOTATATE + 50 mCi (1.85 GBq) [^90^Y]Y-DOTATATE was given to 10 patients. Ninety percent of patients (n = 36) underwent a full four-course-cycle of treatment. Intervals lasting 8 to 14 weeks were maintained, during which long-lasting somatostatin analogues (octreotide 30 mg (16 patients = 40%) or lanreotide 120 mg (24 patients = 60%)) were given every 4 weeks. Every single course of the treatment involved intravenous administration of amino acid nephroprotection (Nephrotec^®^ 100 mg/mL by Fresenius Kabi, Bad Homburg, Germany) before (1000 mL) and one day after (500 mL) radioligand infusion. Biochemical and ultrasound parameters were assessed during course I, course IV and a year after course IV (18 months after the start of RLT).

### 4.2. Inclusion and Exclusion Criteria

The selection of patients qualified for treatment was carried out by joint nationwide endocrinology and oncology councils. All patients involved in the study group were over 18 years old and signed an informed consent form. They had histological confirmation of NEN, with good expression of somatostatin receptors in somatostatin receptor imaging (SRI) performed up to 3 months before therapy started. The morphological presence of a tumor confirmed by computed tomography (CT) or magnetic resonance imaging (MRI) was also necessary.

The detailed inclusion criteria were unresectable metastatic progressive neuroendocrine neoplasm (defined as tumor grade 1 or 2 with Ki-67 < 20%, progression according to the RECIST 1.1 (Response Evaluation Criteria In Solid Tumors) criteria, over the previous 12 months) and a good expression of somatostatin receptors in qualifying somatostatin receptor scintigraphy (SRS) with ^99m^Tc-HYNIC-TOC (combined with SPECT/CT, where radiotracer uptake in the majority of the lesions was higher than in normal liver) or in Gallium-68-PET/CT (SUV_max_ in the majority of the lesions higher than SUV_max_ in normal liver). The exclusion criteria were non-consent, pregnancy or lactation, Karnofsk’y scale < 60, WHO/ECOG 3 or 4, no tracer uptake in SRI, myelosuppression (hemoglobin < 8 g/L, platelets < 80,000/µL, leukocytes < 2000/µL, lymphocytes < 500/µL or neutrophils < 1000/µL), renal dysfunction (estimated glomerular filtration rate (eGFR) < 30 mL/min or serum Creatinine > 1.8 mg/dL) and liver disease (ALT 3× over upper limit) [2].

### 4.3. Laboratory and Ultrasound Evaluation

Venous blood was collected with BD Vacutainer Tests in the Endocrinology and Radioisotope Therapy Department, Military Institute of Medicine, National Research Institute. Samples were analyzed in the Laboratory Diagnostics Department, Military Institute of Medicine, National Research Institute. Analyses were performed on an automatic biochemistry analyzer Cobas C501 (2016) from Roche Diagnostics, Switzerland. Standard renal parameters such as creatinine (Cr; Norm: 0.6–1.2 mg/dL) and GFR (measured with use of CKD-EPI formula, mL/min/1.73 m^2^ [62]) were further extended to analyze the full treatment influence on renal function. Thus, the albuminuria and albumin/creatinine ratio (ACR) were chosen as sensitive markers of renal filtration barrier injury and endothelial damage. Next, the parameters of tubular cell injury and dysfunction were analyzed: fractional excretion of natrium (FE_Na_), potassium (FE_K_), calcium (FE_Ca_), phosphates (FE_PO4_), urea (FE_U_), uric acid (FE_UA_) and urine pH. Moreover, Kidney Injury Molecule 1 (KIM-1; N: 31.2–2000 pg/mL, sensitivity < 2 pg/mL) and Interleukin-18 (IL-18; N: 39–2500 pg/mL; sensitivity < 20 pg/mL) were measured with use of Human KIM-1 ELISA and Human IL-18 ELISA kits from Biorbyt, United Kingdom, to determine potential renal tubular and interstitial radiotoxic and inflammatory injury. These factors are undetermined in NEN patients.

Dynamic tissue perfusion measurements (DTPMs) in the renal cortex were also performed. DTPM is a method of measuring tissue microvascular perfusion with the use of color Doppler sonographic images and specialized software to assess the microvascularity of the organ [63,64]. The renal resistive index (rRI) was estimated in three segmental arteries of each kidney and then averaged. An ultrasound examination was performed by one nephrologist with over 10 years of experience in ultrasound imaging using a Logiq P6 (GE Healthcare, Seoul, Republic of Korea) device equipped with a 2–5 MHz convex transducer. At the time of each ultrasound examination, the blood pressure was measured on the left arm with the use of an Omron 705-IT (Omron Corporation, Kyoto, Japan) device.

### 4.4. Statistical Analysis

Statistical analyses of the results were performed with the IBM SPSS Statistics package, Version 25.0., Armonk, NY, USA: IBM Corp. (Released 2021). Analyses of basic descriptive statistics were performed using the Shapiro–Wilk test and two-way mixed analysis of variance. Differences between dependent variables were analyzed with the use of the appropriate *t*-test or the Wilcoxon test. Differences between groups were analyzed with the appropriate *t*-test or the Mann–Whitney U test. Correlations between variables were analyzed using Pearson’s or Spearman’s test. Missing data were deleted pairwise. A *p*-value of <0.05 was assumed as the level of significance.

## 5. Conclusions

Our study showed that RLT may lead to a 10% decrease in GFR one year after therapy, with a parallel increase in diastolic blood pressure. The new renal parameters, KIM-1 and IL-18, did not appear to be good candidates for markers of RLT nephrotoxicity. Although the renal endothelium was not severely affected by RLT, the renal tubules may be injured. A good biomarker for renal tubule dysfunction after RLT could be the fractional calcium excretion. DTPM appeared to be a method of possible utility in evaluating renal complications during and after RLT; however, the results need to be confirmed in further studies.

## Figures and Tables

**Figure 1 ijms-24-07508-f001:**
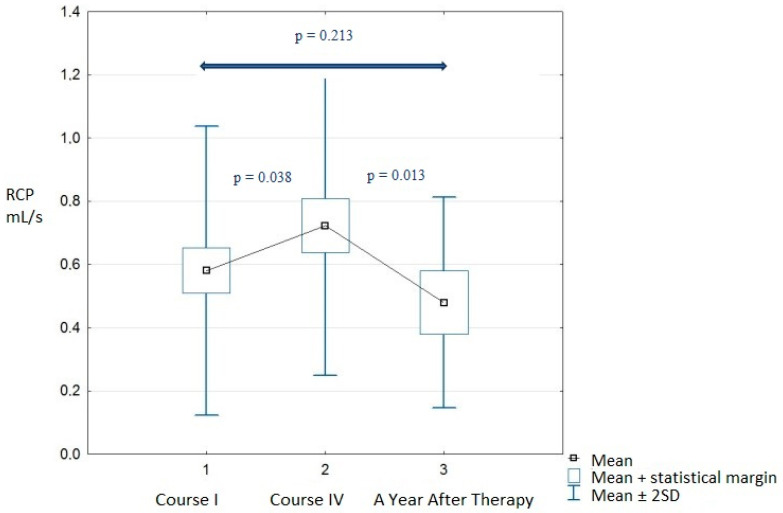
Renal cortex perfusion during observations.

**Figure 2 ijms-24-07508-f002:**
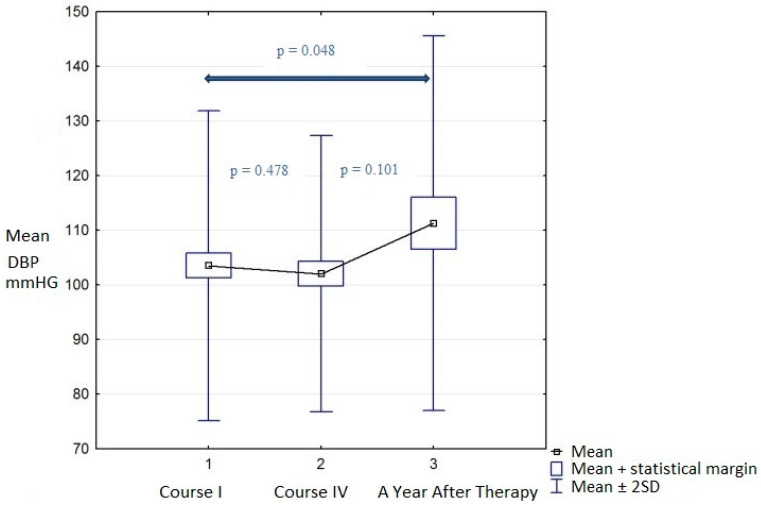
Diastolic blood pressure during observations.

**Table 1 ijms-24-07508-t001:** Characteristics of the study group.

		N = 40	%
Age	mean	58 ± 13	N/A
	range	23–76	N/A
Gender	females	18	45%
	males	22	55%
Primary NEN location	pancreas	15	37.5%
	small intestine	12	30%
	large intestine	5	12.5%
	other (lungs, ovaries, stomach)	4	10%
	unknown	4	10%
NEN Staging	G1	19	47.5%
	G2	21	52.5%
BMI	mean	25 ± 5	N/A
	range	16.4–41.3	N/A
	<18.5	3	7.5%
	28.5–24.9	20	50%
	24.9–29.9	11	27.5%
	≥30.0	6	15%

BMI—body mass index, NEN—neuroendocrine neoplasm; N—number of patients, %—percentage, N/A—not applicable.

**Table 2 ijms-24-07508-t002:** Renal parameters before and after courses I and IV.

Renal Parameters	Unit	Before Course I (n = 40)		After Course I (n = 40)		*p*	Before Course IV (n = 35)		After Course IV (n = 35)		*p*
		M	SD	M	SD		M	SD	M	SD	
Cr	mg/dL	0.93	0.29	0.93	0.29	0.317	0.93	0.3	0.96	0.3	0.161
GFR	mL/min/1.73 m^2^	85.23	25.05	85.15	25.96	0.290	87.81	26.32	84.74	24.00	0.205
FE_Na_	%	0.88	1.20	0.94	0.58	0.291	0.66	0.41	0.77	0.49	0.356
FE_K_	%	10.48	7.8	6.53	3.55	0.044	10.10	4.21	16.20	14.18	0.111
FE_Ca_	%	0.58	0.57	1.87	5.91	0.320	0.34	0.31	0.55	0.49	0.046
FE_PO4_	%	12.50	7.84	15.00	7.18	0.098	12.47	6.27	15.09	6.57	0.223
FE_U_	%	35.17	20.65	50.66	16.36	<0.001	36.97	10.55	49.40	9.47	<0.001
FE_UA_	%	5.61	0.84	5.66	0.60	0.395	6.52	3.86	5.89	3.13	0.294
ACR	mg/g	0.08	0.15	0.08	0.31	0.527	0.02	0.03	0.01	0.01	0.003
Urine albumins	mg/mL	9.47	23.45	1.12	2.64	0.004	2.96	5.77	0.67	0.90	0.035
Serum albumins	mg/dL	4.54	0.52	4.28	0.55	<0.001	4.57	0.33	4.37	0.33	<0.001
Urine pH	NA	6.75	5.11	7.39	3.94	0.047	5.52	0.62	5.52	0.53	0.747
Urine KIM-1	pg/dL	1906.33	1469.74	1424.52	1482.91	0.147	1456.37	1380.13	1307.38	1176.87	0.593
Urine IL-18	pg/dL	160.84	124.47	46.78	51.7	<0.001	53.51	61.37	92.16	417.43	0.090
Serum KIM-1	pg/mL	74.32	282.96	65.47	197.17	0.987	24.87	92.73	19.66	57.34	0.751

M—mean value; SD—standard deviation; GFR—glomerular filtration rate; Cr—creatinine; FE—fractional excretion, Na—sodium, K—potassium, Ca—calcium, PO4—phosphates, U—urea, UA—uric acid; ACR—albumin/creatinine ratio; KIM-1—kidney injury molecule 1; IL-18—interleukin 18.

**Table 3 ijms-24-07508-t003:** Renal parameters before course I and before course IV.

Renal Parameters	U	Change during Course I (n = 33)		Change during Course IV (n = 33)		*p*
		Δ	SD	Δ	SD	
Cr	mg/dL	−0.03	0.17	−0.003	0.3	0.526
GFR	mL/min/1.73 m^2^	−0.68	12.07	−2.79	7.98	0.036
FE_Na_	%	0.06	1.02	0.11	0.63	0.733
FE_K_	%	−4.39	7.73	6.86	14.43	0.012
FE_Ca_	%	2.43	8.30	2.63	5.91	0.894
FE_PO4_	%	1.37	6.38	0.22	0.49	0.590
FE_U_	%	13.80	24.57	12.82	12.57	0.646
FE_UA_	%	0.03	4.02	−0.63	2.82	0.154
ACR	mg/g	0.01	0.22	−0.01	0.02	0.845
Urine albumins	mg/mL	−6.93	19.26	−2.33	5.09	0.020
Serum albumins	mg/dL	−0.26	0.33	−0.21	0.26	0.656
Urine pH	N/A	0.03	0.87	0.00	0.77	0.316
Urine KIM-1	pg/dL	−313.66	1511.83	−209.34	1644.21	0.326
Urine IL-18	pg/dL	−116.5	129.10	−30.39	57.03	<0.001
Serum KIM-1	pg/mL	−10.85	108.16	−3.41	42.76	0.998

U—units; Δ—change; SD—standard deviation; GFR—glomerular filtration rate; Cr—creatinine; FE—fractional excretion, Na—sodium, K—potassium, Ca—calcium, PO4—phosphates, U—urea, UA—uric acid; ACR—albumin/creatinine ratio; KIM-1—kidney injury molecule 1; IL-18—interleukin 18; N/A—not applicable.

**Table 4 ijms-24-07508-t004:** Analysis of acute complications in patients treated with Lu vs. Lu/Y (during course I).

	[^177^Lu]Lu-DOTATATE (n = 30)		[^177^Lu]/[^90^Y]YLu-DOTATATE (n = 10)		*p*
	Δ	SD	Δ	SD	
GFR CKD-EPI (mL/min/1.73 m^2^)	−2.62	10.64	6.64	10.38	0.018
FE_K_ (%)	−5.08	8.40	−0.69	3.76	0.121
FE_U_ (%)	13.28	26.39	21.90	13.47	0.332
Urine Albumins (mg/mL)	−4.25	17.19	−20.65	33.71	0.169
Urine IL-18 (pg/dL)	−104.06	131.54	−130.47	79.81	0.555

Δ—change (before and after course I); SD—standard deviation; GFR CKD-EPI—glomerular filtration rate Chronic Kidney Disease Epidemiology Collaboration; FE—fractional excretion, K—potassium, U—urea; IL-18—interleukin 18.

**Table 5 ijms-24-07508-t005:** Summary of renal parameters at different times of evaluation.

Parameters	U	Before Course I (n = 36)		Before Course IV (n = 36)		*p*	Before Course IV (n = 19)		A Year after Course IV (n = 19)		*p*	Before Course I (n = 19)		A Year after Course IV (n = 19)		*p*
		M	SD	M	SD		M	SD	M	SD		M	SD	M	SD	
Creatinine	(mg/dL)	0.91	0.28	0.92	0.31	0.388	0.95	0.35	1.04	0.42	0.155	0.92	0.27	1.04	0.42	0.036
GFR	(mL/min/1.73 m^2^)	86.57	26.33	84.74	24.00	0.930	87.74	30.82	80.95	32.25	0.06	88.11	26.47	80.95	32.25	0.009
FE_Na_	(%)	0.82	1.11	0.67	0.42	0.947	0.66	0.47	0.71	0.49	0.708	0.65	0.69	0.71	0.49	0.927
FE_K_	(%)	10.24	7.55	10.03	4.33	0.033	10.73	4.80	10.32	4.91	0.790	8.56	4.69	10.32	4.91	0.669
FE_Ca_	(%)	0.52	0.45	0.34	0.32	0.306	0.29	0.23	3.12	2.17	0.003	0.56	0.48	3.12	2.17	0.009
FE_PO4_	(%)	12.49	8.11	12.7	6.47	0.802	13.99	9.05	14.21	7.07	0.437	13.99	9.05	14.21	7.07	0.437
FE_U_	(%)	35.31	21.55	36.52	10.66	0.536	36.74	11.57	38.73	11.03	0.710	36.82	19.69	38.73	11/03	0.836
FE_UA_	(%)	7.12	5.44	6.58	3.95	0.843	5.82	2.41	6.64	3.24	0.457	7.12	5.13	6.64	3.24	0.561
ACR	mg/g	0.07	0.13	0.03	0.03	0.079	0.02	0.02	0.10	0.29	0.654	0.07	0.16	0.13	0.34	0.803
Urine albumins	(mg/mL)	7.84	19.14	2.89	5.79	0.011	1.96	2.83	4.80	8.46	0.147	2.97	4.37	4.80	8.46	0.314
Serum albumins	(mg/dL)	4.61	0.39	4.58	0.33	0.564	4.60	0.42	4.51	0.40	0.172	4.61	0.43	4.51	0.40	0.712
Urine pH	NA	5.55	0.82	5.53	0.62	0.386	5.62	0.65	5.5	0.59	0.807	5.5	0.66	5.5	0.59	0.631
urine KIM-1	(pg/mL)	1851.1	1344	1416.1	1393.1	0.003	1839	2443	934.6	616	0.564	1799.83	1426.41	934.65	615.98	0.155
urine IL-18	(pg/mL)	167.3	126.5	47.94	58.1	<0.001	65.95	73.18	37.26	18.16	0.856	183.8	145.7	37.26	18.16	0.084
Serum KIM-1	(pg/mL)	67.75	291.5	22.19	91.19	0.545	5.73	16.19	4.00	7.71	0.309	24.8	63/93	4.00	7.71	0.931

U—units; M—mean value; SD—standard deviation; GFR—glomerular filtration rate; FE—fractional excretion, Na—sodium, K—potassium, Ca—calcium, PO4—phosphates, U—urea, UA—uric acid; ACR—albumin/creatinine ratio, KIM-1—kidney injury molecule 1; IL-18—interleukin 18.

**Table 6 ijms-24-07508-t006:** Combined dynamic tissue perfusion measurement (DTPM) parameters throughout the study.

DTPM	U	Before Course I (n = 31)		Before Course IV (n = 31)		*p*	Before Course IV (n = 11)		Year after Therapy (n = 11)		*p*	Before Course I (n = 11)		One Year after Therapy (n = 11)		*p*
		M	SD	M	SD		M	SD	M	SD		M	SD	M	SD	
Vmean	cm/s	1.815	0.640	1.946	0.646	0.079	1.782	0.529	1.566	0.603	0.090	1.800	0.661	1.58	0.551	0.213
cRI	ratio	0.699	0.132	0.691	0.117	0.740	0.727	0.143	0.722	0.164	0.880	0.688	0.111	0.706	0.151	0.606
VA	cm^2^	0.283	0.157	0.340	0.152	0.031	0.350	0.162	0.253	0.138	0.025	0.322	0.164	0.291	0.152	0.397
RCP	mL/s	0.544	0.389	0.686	0.436	0.038	0.677	0.456	0.427	0.350	0.013	0.579	0.372	0.479	0.334	0.213
rRI	ratio	0.668	0.061	0.675	0.053	0.413	0.684	0.064	0.682	0.066	0.821	0.683	0.070	0.682	0.061	0.953
SBP	mmHg	142.85	23.98	140.00	22.66	0.307	140.82	20.99	149.27	26.97	0.147	144.25	28.999	152.667	27.364	0.216
DBP	mmHg	84.74	12.51	83.33	9.43	0.480	83.27	9.825	87.55	12.49	0.098	83.167	13.523	90.833	14.503	0.048
MAP	mmHg	104.11	15.55	102.22	13.36	0.478	102.46	12.65	108.12	16.59	0.101	103.528	18.116	111.444	17.903	0.091
PP	mmHg	58.11	15.61	56.67	15.39	0.636	57.55	14.15	61.73	17.90	0.314	61.083	18.253	61.833	17.658	0.848
HR	1/min	75.07	10.91	71.93	11.56	0.081	75.73	10.09	74.82	14.84	0.762	77.00	9.254	76.167	14.733	0.768

DTPM—dynamic tissue perfusion measurement, U—units, Vmean—mean velocity, cRI—cortex resistance index, VA—vascular area, RCP—renal cortex perfusion, rRI—resistive index in renal segmental artery, SBP—systolic blood pressure, DBP—diastolic blood pressure, MAP—mean arterial pressure, PP—pulse pressure, HR—hearth rate; M—mean, SD—standard deviation, *p*—*p*-value,

**Table 7 ijms-24-07508-t007:** GFR calculation using different methods of estimation.

	Before Course IV (n = 19)		A Year after Treatment (n = 19)		*p*
	M	SD	M	SD	
GFR CKD-EPI cr (mL/min/1.73 m^2^)	87.74	30.82	80.95	32.25	0.060
GFR MDRD (mL/min/1.73 m^2^)	87.00	33.7	78.32	32.82	0.024

M—mean value; SD—standard deviation; GFR CKD-EPI—glomerular filtration rate Chronic Kidney Disease Epidemiology Collaboration; GFR MDRD—glomerular filtration rate Modification of Diet in Renal Disease.

**Table 8 ijms-24-07508-t008:** Analysis of chronic complications in patients treated with ^177^Lu vs. ^177^Lu/^90^Y.

	Before Course IV and a Year after Therapy					Before Course I and a Year after Therapy				
Parameter	[^177^Lu]Lu-DOTATATE (n = 16)		[^177^Lu]/[^90^Y]YLu-DOTATATE (n = 11)		*p*	[^177^Lu]Lu-DOTATATE (n = 16)		[^177^Lu]/[^90^Y]YLu-DOTATATE (n = 11)		*p*
	Δ	SD	Δ	SD		Δ	SD	Δ	SD	
GFR CKD-EPI (mL/min/1.73 m^2^)	−5.94	14.07	−11.33	9.71	0.538	−5.63	10.83	−15.33	14.50	0.191
FE_Ca_ (%)	9.04	0.40	9.62	0.23	0.029	8.81	0.72	9.50	0.37	0.128

Δ—change (Δ before course IV and a year after therapy or before course I and a year after therapy); SD—standard deviation; GFR CKD-EPI—glomerular filtration rate Chronic Kidney Disease Epidemiology Collaboration; FE_Ca_—fractional calcium excretion.

## Data Availability

All appropriate and significant data had been placed in the manuscript. Due to privacy and ethical reasons, some anonymized data can be presented after written request.

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
