# Peer review of "Renal Disturbances during and after Radioligand Therapy of Neuroendocrine Tumors—Extended Analysis of Potential Acute and Chronic Complications"

_ijms, 2023, doi:10.3390/ijms24087508_

Round 1

Reviewer 1 Report

The authors designed a study of great interest, presenting an interesting and still underresearched subject that may have a major impact in the management of patients treated with PRRT. However, there are some key problems that need to be addressed, in order to improve the quality of the manuscript. My suggestions are as it follows:

  1. Firstly, the introduction of the article is too short and provides little background for the study purpose. I suggest adding more information about PRRT and how it could influence the renal function and its renal adverse effects, as well as some details about the novel biomarkers used to address the renal function impairment (such as IL-18 and KIM-1, mentioned in the text);
  2. Please specify which amino-acids had been used for nephroprotection, as mentioned in the first section of M&M.
  3. The mentioned patient inclusion criteria are those required for PRRT. However, the authors should give more details about the patient selection procedure for their study (such as patients over 18 years old, who accepted the procedure etc.)
  4. Which was the radiotracer used for SRS, In111 or Tc99m? We know that the sensibility and specificity of the two might differ.
  5. The authors should also add the information about the performed statistical analysis in a separate section in “Materials and Methods” category.
  6. Please specify how many patients were in the non-diabetes subgroup, analysed for acute complications during the fourth course of RTL.
  7. I suggest adding a separate table comparing the results obtained for the only Lu177 group with the ones obtained for Lu177/Y90, as the authors mention throughout the study that the combination of the radiotracers may be more nephrotoxic.
  8. I believe that Analysis One from Chronic complications coincides with the analysis of acute complications. However, the parameters found as statistically significant differ between the two, as there are less patients. Actually, an important issue of the study is that the number of patients varies a lot between and, while this affects the statistics, it is specified nowhere in the text, but only visible on the Tables. I do believe that, for comparison purposes, the numbers in the subgroups should be equal (as done in table 8).
  9. Table 6 does not exist.
  10. Again, when comparing the diastolic blood pressure (DBP), the authors compare a mean DBP obtained from 40 patients to a one obtained for 11 patients, which might not convey into truthful results. I suggest making these comparisons only on the 11 patients that have been evaluated at least twice (same as for Analysis Two, table 7, where the authors compare, among others, the DTPM).
  11. I suggest adding a separate table with the compared results obtained for Lu177 group and Lu177/Y90 group in the Chronic complications as well.
  12. Were dosimetry studies of the renal radiotracer absorbed dose performed? Have the results correlated with the ones obtained for the urinary and serum biomarkers?
  13. Study limitations need to be improved as well, as not the number of patients represents a problem, but the number of individuals who presented for follow-up one year after therapy.
  14. I suggest reformulating the conclusions in a more shorter paragraph, as the information is found throughout the text.
  15. There is a need for a thorough assessment of grammar errors and English language, as there are many mistakes found throughout the manuscript.

I am once again reiterating the great idea of the study, but these changes need to be performed in order to provide a qualitative manuscript.

Author Response

Dear Reviewer,

First of all, we would kindly like to thank you for the review. We have corrected the manuscript according to all your valuable comments and suggestions. We hope the corrected manuscript will meet your all expectations. Below, we have attached the answers for all of your questions.

  1. Firstly, the introduction of the article is too short and provides little background for the study purpose. I suggest adding more information about PRRT and how it could influence the renal function and its renal adverse effects, as well as some details about the novel biomarkers used to address the renal function impairment (such as IL-18 and KIM-1, mentioned in the text);

Introduction updated.

  1. Please specify which amino-acids had been used for nephroprotection, as mentioned in the first section of M&M.

The amino-acids that had been used for nephroprotection was Nephrotec® 100mg/mL (Fresenius Kabi). Manuscript updated.

  1. The mentioned patient inclusion criteria are those required for PRRT. However, the authors should give more details about the patient selection procedure for their study (such as patients over 18 years old, who accepted the procedure etc.)

The selection of patients qualified for treatment was carried out on joint nationwide endocrinology and oncology councils, and only patients meeting PRRT criteria were included into the study. However, we have updated the manuscript in some details according to your suggestions..

  1. Which was the radiotracer used for SRS, In111 or Tc99m? We know that the sensibility and specificity of the two might differ.

The radiotracer used for SRS was 99mTc. Manuscript updated.

  1. The authors should also add the information about the performed statistical analysis in a separate section in “Materials and Methods” category.

Manuscript updated.

  1. Please specify how many patients were in the non-diabetes subgroup, analysed for acute complications during the fourth course of RTL.

The number of non-diabetes patients was 12.

  1. I suggest adding a separate table comparing the results obtained for the only Lu177 group with the ones obtained for Lu177/Y90, as the authors mention throughout the study that the combination of the radiotracers may be more nephrotoxic.

Tables prepared. Manuscript updated.

  1. I believe that Analysis One from Chronic complications coincides with the analysis of acute complications. However, the parameters found as statistically significant differ between the two, as there are less patients. Actually, an important issue of the study is that the number of patients varies a lot between and, while this affects the statistics, it is specified nowhere in the text, but only visible on the Tables. I do believe that, for comparison purposes, the numbers in the subgroups should be equal (as done in table 8).

The “acute” assessment is one, where parameters were measured separately during course I and course IV of treatment (i.e. before and after course I or before and after course IV). “Chronic” assessment is a comparison of parameters before I, before IV and one year after therapy. Differences in the size of the groups in the individual phases of the study result from the natural process of decreasing the number of patients for various reasons (which are described in the study limitations section) during the time of observation. After discussion with our biostatisticians, we decided to analyze each stage separately (so that not to lose data from previous stages), hence the differences in the size of the groups at different stages of the study.

  1. Table 6 does not exist.

Tables numeration updated.

  1. Again, when comparing the diastolic blood pressure (DBP), the authors compare a mean DBP obtained from 40 patients to a one obtained for 11 patients, which might not convey into truthful results. I suggest making these comparisons only on the 11 patients that have been evaluated at least twice (same as for Analysis Two, table 7, where the authors compare, among others, the DTPM).

We would like to kindly apologize for the mistake in patient group numeration. The proper the size of the group on which the analysis was based is 11 for both subgroups.

  1. I suggest adding a separate table with the compared results obtained for Lu177 group and Lu177/Y90 group in the Chronic complications as well.

Manuscript updated

  1. Were dosimetry studies of the renal radiotracer absorbed dose performed? Have the results correlated with the ones obtained for the urinary and serum biomarkers?

Unfortunately, the dosimetry were not planed and performed in this study, so we were not able to correlate those results. The study was planned in 2016-2017 when dosimetry were not mandatory for radioisotopes therapy.

  1. Study limitations need to be improved as well, as not the number of patients represents a problem, but the number of individuals who presented for follow-up one year after therapy.

Manuscript updated.

  1. I suggest reformulating the conclusions in a more shorter paragraph, as the information is found throughout the text.

Manuscript updated. Conclusions corrected.

There is a need for a thorough assessment of grammar errors and English language, as there are many mistakes found throughout the manuscript.

The manuscript was edited.

Reviewer 2 Report

The study seems fine, methods are written in detail, data are convincing, article language is ok throughout the manuscript. There are grammatical mistakes in the manuscript. Sentence in the 1st paragraphs in exclusion /inclusion criteria is too long and loses its meaning. There are few mistakes which can be easily corrected such as:

Introduction: line 8- ‘also requires’

 The study by Marek Saracyn et al. has clinical relevance in treating patients who underwent RLT treatment which caused glucose metabolism disruption with increase of fasting glucose concentration. Treating such patients appropriately may prevent further complications in patients who are already dealing with RLT adverse effects.

 The introduction part lacks the explanation of mechanism of glucose metabolism which seems the main topic of the manuscripts.

It's understandable that the sample size of the research was low due to low incidence of neuroendocrine neoplasms in population.

Authors mentioned of lack of complete glucose metabolism parameters (HbA1c, insulin, or c-peptide) and insulin resistance factors (like HOMA-IR or Matsuda index) in all patients as a limitation seems unreasonable- in a study which is determining glucose metabolism.

 The reasoning and explanation to prove hypothesis of the manuscript can be improved, which will further improve the quality of the manuscript.

Author Response

Dear Reviewer,

We would like to kindly apologize for the mistake. During process of submission a wrong file was uploaded, and unfortunately you had received the version, that was send to another MDPI  Journal.

However, we would like to thank you for the review – your suggestions helped us to improve the other manuscript.

Best regards

Authors Team

Reviewer 3 Report

1. Is the change in glucose clinical relevant?

2. How many glucose levels did you measure before and after PRRT treatments? And are they fasting levels or random glucose measurements?

3. Did you consider measuring postprandial glucose, day profile or putting on a CGM or isCGM?

4. If you have fasting glucose and insulin why not calculate HOMA-IR?

5. How many developed diabetes after PRRT?

6. How did HbA1c change during and 1 year after treatment?

Author Response

(The authors gave the same response as above.)

Reviewer 4 Report

While the topic of radioligand therapy in neuroendocrine tumors is certainly of interest to the medical community, I did not find that the manuscript presented sufficiently novel findings or insights to justify publication in IJMS journal. The research methodology appears to be standard, and the sample size is relatively small. The study does not present any groundbreaking or unique discoveries that would make it a valuable addition to the scientific literature. In addition, the introduction section of the manuscript is not comprehensive enough to provide the necessary background information on the topic of neuroendocrine tumors and radioligand therapy. The introduction is a crucial section of the manuscript, and it is essential that it presents a clear and concise overview of the research area, the key concepts, and the research question or hypothesis. However, in this manuscript, the introduction is lacking in several areas, including a detailed overview of neuroendocrine tumors and their treatment options, and a thorough review of existing literature on radioligand therapy. Furthermore, the conclusion section is not concise and fails to effectively summarize the main findings of the research. A conclusion should provide a clear and concise summary of the research, highlighting the key findings and their significance. Overall, I recommend rejecting this manuscript for publication in IJMS journal. However, I encourage the authors to revise the manuscript to address the issues mentioned above and consider submitting it to a more specialized journal in the field.

Author Response

First of all, we would kindly like to thank you for the review. We have corrected the manuscript according to all your valuable comments and suggestions. We hope the corrected manuscript will meet your all expectations.

Round 2

Reviewer 1 Report

Thank you very much for your prompt responses.

The authors performed the changes according to my suggestions. I would still recommend reformulating the conclusions to sound more like a concluding paragraph rather than a short resume, but other than that, the manuscript is in a good form for publication.

Author Response

Dear Reviewer,

We have reformulated the conclusions paragraph. We would also kindly thank you for the review and recommendation.

Best regards,

Reviewer 4 Report

While the author has made significant improvements based on my feedback, I believe that the conclusion section still requires further revision.

Author Response

Dear Reviewer

We have revised the conclusions section. We hope the updated manuscript will meet all of your expectations.

Best regards,